# An integrative, multi-scale, genome-wide model reveals the phenotypic landscape of *Escherichia coli*

Javier Carrera[1,†], Raissa Estrela[2], Jing Luo[1], Navneet Rai[1], Athanasios Tsoukalas[1,3] & Ilias Tagkopoulos[1,3,*]

## Abstract

Given the vast behavioral repertoire and biological complexity of even the simplest organisms, accurately predicting phenotypes in novel environments and unveiling their biological organization is a challenging endeavor. Here, we present an integrative modeling methodology that unifies under a common framework the various biological processes and their interactions across multiple layers. We trained this methodology on an extensive normalized compendium for the gram-negative bacterium *Escherichia coli,* which incorporates gene expression data for genetic and environmental perturbations, transcriptional regulation, signal transduction, and metabolic pathways, as well as growth measurements. Comparison with measured growth and high-throughput data demonstrates the enhanced ability of the integrative model to predict phenotypic outcomes in various environmental and genetic conditions, even in cases where their underlying functions are underrepresented in the training set. This work paves the way toward integrative techniques that extract knowledge from a variety of biological data to achieve more than the sum of their parts in the context of prediction, analysis, and redesign of biological systems.

**Keywords** genome engineering; genome-scale model; model-driven experimentation; predictive modeling and integration; systems and synthetic biology
**Subject Categories** Computational Biology; Genome-Scale & Integrative Biology; Metabolism
**Mol Syst Biol.** (2014) 10: 735

## Introduction

The development of an integrative genome-scale model is considered to be the Holy Grail of computational predictive modeling in systems biology (Tomita, 2001). The potential of such a feat is transformative and spans most areas of life science research: discovery of novel properties and emerging behaviors at the organism level,

generating and testing predictable hypotheses in well-defined simulated environments, guiding experimentation, and accelerating the in-depth understanding of cellular physiology. Despite its utility, whole-cell modeling across multiple scales remains elusive due to a number of factors. First, even for well-studied organisms, we still have a limited knowledge of the cellular machinery, pathways, proteins, and their respective functions (Frishman, 2007; Hanson *et al*, 2010). Furthermore, the complex interconnectivity and interdependencies of cellular processes render their detailed mapping a challenging task that is further hindered by the lack of comprehensive quantitative data across different environmental conditions. The latter is rapidly changing, however, due to the technological advances in high-throughput sequencing that enable the acquisition of an unprecedented amount of data that span all aspects of cellular organization. Concomitantly, research advances in the computational front have reached the level of maturity needed for the analysis and integration of these datasets.

Early work in integrative modeling under one umbrella was E-cell (Tomita *et al*, 1999), a modular software environment for whole-cell simulation that included organelle sub-models (Yugi & Tomita, 2004). More recently, genome-scale simulations were performed to study complex phenomena, such as the emergence of anticipatory behavior during evolution in varying environments (Tagkopoulos *et al*, 2008), the noise contributions of an inducible switch (Roberts *et al*, 2011) and the effect of stochastic expression to metabolic variability (Labhsetwar *et al*, 2013). A whole-cell model of *Mycoplasma genitalium*, a human urogenital parasite whose genome contains 525 genes and is described by 28 cellular processes, was presented recently with encouraging results on the prediction of cellular behavior (Karr *et al*, 2012). A crucial tool for integrative modeling is network inference algorithms, both unsupervised and supervised, which can be used to generate topological models and consensus networks from data (Basso *et al*, 2005; Faith *et al*, 2007; Mordelet & Vert, 2008; Taylor *et al*, 2008; Zare *et al*, 2009; Marbach *et al*, 2010, 2012). Several methods have targeted the integration of models across the transcriptional, proteomic, signal transduction, and metabolomics layers (Reed *et al*, 2003, 2006; Covert *et al*, 2004; Duarte *et al*, 2004; Beltran *et al*, 2006; Joyce & Palsson,

1   UC Davis Genome Center, University of California, Davis, CA, USA
2   Department of Molecular and Cell Biology, University of California, Berkeley, CA, USA
3   Department of Computer Science, University of California, Davis, CA, USA
    *Corresponding author. Tel: +1 530 752 7707; Fax: +1 530 752 4767; E-mail: iliast@ucdavis.edu
    †Present address Department of Bioengineering, Stanford University, Stanford, CA, USA

2006; Kresnowati *et al*, 2006; Becker *et al*, 2007; Feist *et al*, 2007; Andersen *et al*, 2008; Feist & Palsson, 2008; Herrgard *et al*, 2008; Carrera *et al*, 2012a,b).

Our aim here was to construct a phenomenological model for bacterial organisms that integrates multiple layers of biological organization. We focused on a genome-scale model for *Escherichia coli*, a gram-negative, facultative anaerobic model bacterium. *E. coli* serves as an ideal candidate for multi-scale cell modeling, due to the wealth of data and knowledge accumulated over the years, the easiness to culture and manipulate experimentally, and its importance in medical and biotechnological applications. Figure 1 depicts the training–simulation–refinement methodology that can be used for the construction of data-driven genome-scale models. Starting from a collection of "omics" data (Fig 1A), cellular processes are divided into modules, constructed from composite networks, and data-driven sub-models that are ultimately integrated under a unifying framework (Fig 1B). Parameters are trained so that the model

optimally captures the observed relationships given an objective function and a set of constraints, and the predictive ability of the model is then assessed through a number of statistical tests (Fig 1C). Such a model can be used to generate and test biological hypotheses through simulations pertaining to genetic and environmental perturbations that can subsequently be validated through targeted experimentation (Fig 1D). A critical aspect of any data-driven model is to identify the areas where further experimentation is needed to accurately capture phenomena and biological processes, so that targeted experiments can be performed to address these shortcomings. The resulting experimental data are then integrated to the training dataset, which in turn increase the predictive power of the model.

Toward this goal, we constructed a normalized gene expression (4,189 genes in 2,198 microarrays from 127 scientific articles), signal transduction (151 regulatory pathways, 152 publications), and phenomics (616 arrays) compendium (Fig 2). The constructed

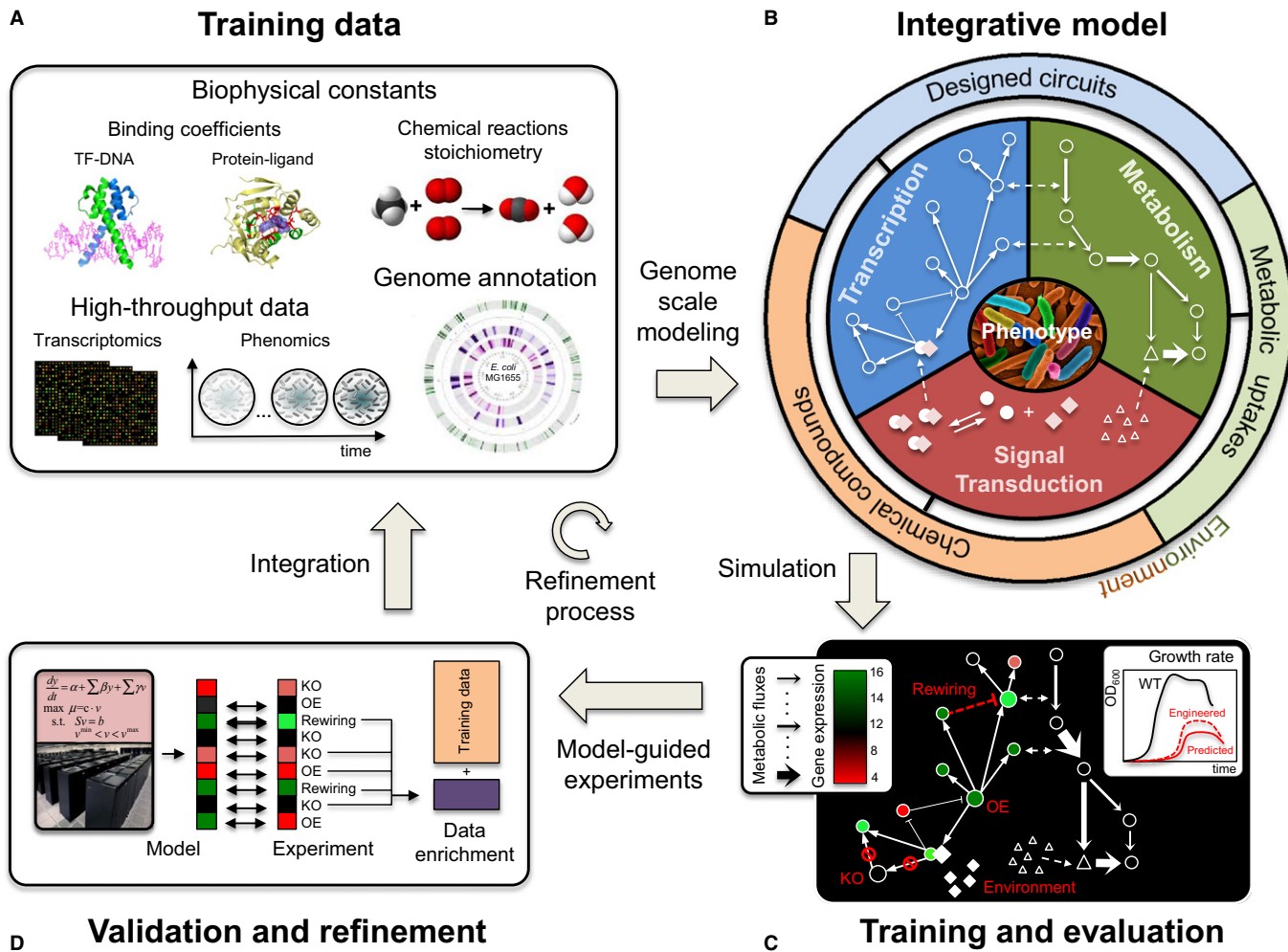

**Figure 1. Overview of integrative modeling through targeted experimentation.**

A   Collection of training data across multiple layers of cellular organization and from various data sources.

B   Development and training of a multi-scale model that integrates transcription, signal transduction and metabolism.

C   Evaluation of model generalization via comparison between predicted and measured growth and expression data.

D   Experimentally test hypotheses generated by the model and incorporate new measurements in the training set.

    

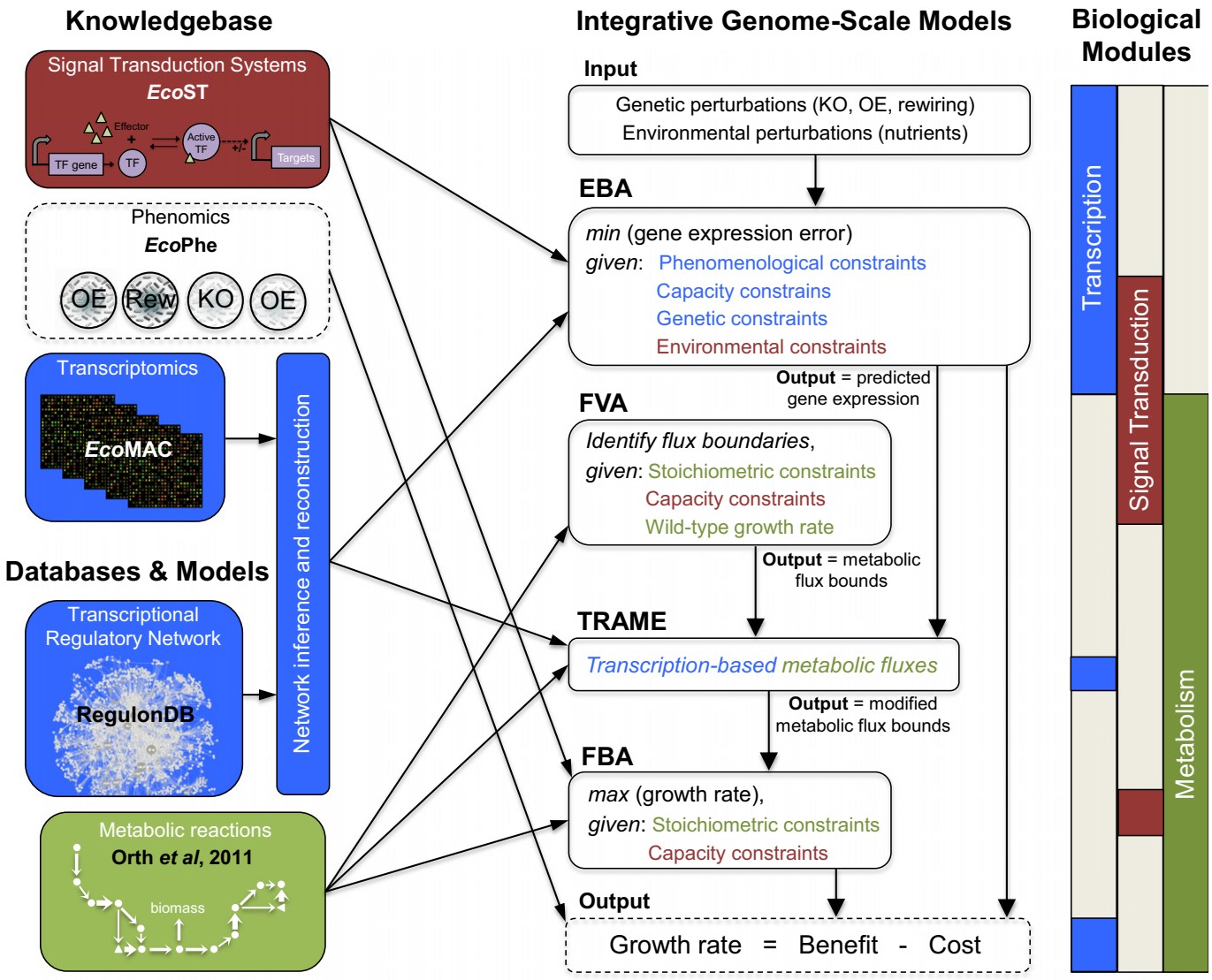

**Figure 2.  Integration of signal transduction, gene expression, metabolic and phenomics data within an integrative framework.**
Links between databases, models, and knowledgebase to the computational methodologies depict the dependences between the various modules in the integrative genome-scale model.

knowledgebase was then integrated with a recently published *E. coli* metabolic model (2,583 reactions and 1,805 metabolites) (Orth *et al*, 2011). The construction of this compendium led to significantly improved predictions by highly ranked inference methods. To allow for genetic and environmental perturbations, we developed a quadratic programming method coined as "*Expression Balance Analysis*" (EBA) that takes into account genetic, capacity, phenomenological, and environmental constraints to predict gene expression. We extended the current models for flux boundary calculations by developing a new method called "TRAnscription-based Metabolic flux Enrichment" (TRAME) that accounts for both metabolic and transcriptional interactions. Statistical tests and subsequent experimental validation demonstrate the capacity of this integrative model to predict environmental and genetic perturbations beyond current stand-alone metabolic and expression (ME) models.

# Results

### Genetic and environmental gene expression diversity

Genetic diversity analysis in *Eco*MAC shows that genetic perturbations led to more diverse gene expression profiles than environmental changes (Kolmogorov–Smirnov test $P < 0.023$; Mann–Whitney test $P < 10^{-15}$; Supplementary Fig S4A and B). In addition, different types of genetic perturbations had a profoundly different expression profile: the gene expression diversity observed in arrays of TF rewiring experiments is more than 2.1-fold ($P < 10^{-10}$) higher than in arrays from single-TF perturbation experiments such as TF knockouts or TF over-expressions. We did not observe significant differences in the variability signatures when comparing arrays of knockouts and over-expression experiments in TFs, enzymes, or other genes. Nonetheless, genetic perturbations of TFs led to

significantly higher expression diversity levels (Mann–Whitney test $P < 10^{-18}$; Kolmogorov–Smirnov test $P < 10^{-17}$) than other genes (Supplementary Fig S4C and D). These results argue that transcriptional rewiring of the existing transcriptional regulatory network (TRN) tends to create larger ripple effects that reverberate across the global transcriptional network, when compared to other single-gene perturbations.

Visualization of the gene targets present in *Eco*MAC reveals a remarkably sparse landscape of genetic and environmental perturbations that have been conducted so far (Fig 3A). Overlap of *Eco*MAC and *Eco*ST depicts clusters of TFs that are implicated in sensing environmental states, such as variations in carbon, nitrogen, and phosphate sources, as well as oxygen, metals, and other supplements (Fig 3B). The calculated effector strength in the whole

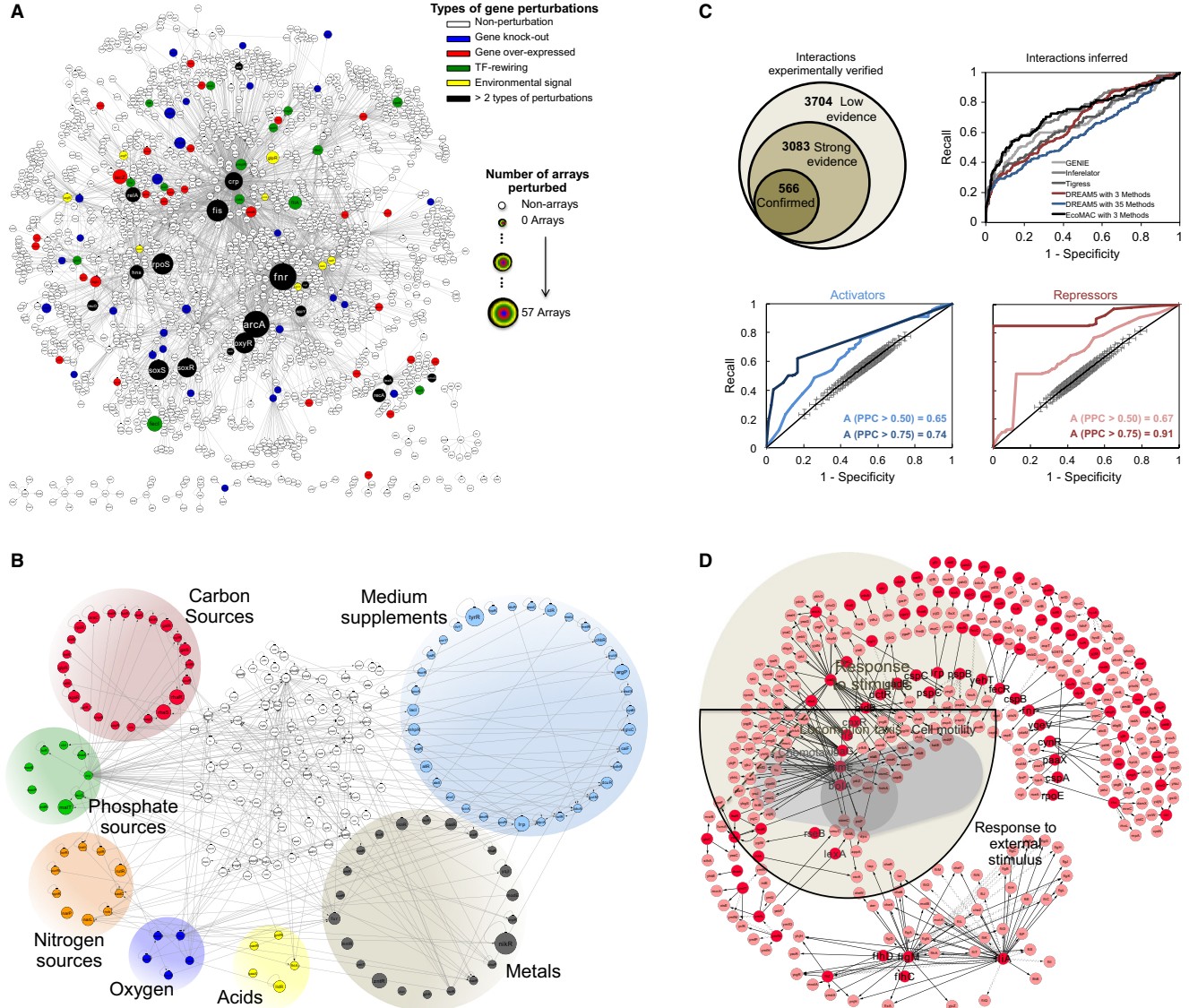

**Figure 3. Map of genetic perturbations, signal transduction pathways and inferred interactions.**

A    TRN of *Escherichia coli* including all transcriptional interactions experimentally verified. Color nodes represent genes identified as genetic perturbations in *Eco*MAC. The TRN contains 1,591 genes (182 TFs representing a 55.5% of the total), and 3,704 transcriptional interactions. 97 of the 141 genetic perturbations fall within the experimentally verified interactions and are shown here.

B    STSs of *Escherichia coli* represented in the *Eco*ST TRN. Nodes represent TFs that are related to carbon sources (red), metals (grey), acids (yellow), nitrogen sources (orange), oxygen (blue), phosphate sources (green), or supplements such as amino acids or precursors of amino acids (light blue). All transcriptional interactions between TFs are represented (416 regulations between 183 TFs).

C    Number of transcriptional interactions with low, strong, and confirmed evidence and ROC/PR curves for predicted transcriptional activators/repressors (bottom panels) and inferred interactions (top, right panel). The performance of three methods trained on *Eco*MAC (AUC = 0.73, AUPRC = 0.23) and 35 methods trained on the 809 arrays of DREAM 5 (AUC = 0.6, AUPRC = 0.23) is shown. The confirmed interactions in RegulonDB v8.1 constitute the golden standard.

D    GO enrichment of the top 500 inferred interactions (0.45 precision threshold).

Source data are available online for this figure.

spectrum of environment-sensing regulatory mechanisms reveals a bias toward highly sensitive TF-effector pairs, where small changes have major implications in cellular expression (Supplementary Fig S9).

### An integrative knowledgebase as a base to regulatory network enrichment

We performed network analysis to tap on *Eco*MAC's potential to reveal novel interactions in *E. coli*'s TRN. We used inference methods that were highly ranked in the latest DREAM challenge to capture distinct *Eco*MAC features by applying mutual information, constrained regression, tree-based methods, and other statistical techniques. Remarkably, inference of regulatory interactions based on the *Eco*MAC compendium increased the performance of the community classifier by 8 to 22%, depending on the number of methods and golden standard used (Supplementary Fig S5). From the top 500 computationally inferred interactions (precision cutoff at 0.45, Fig 3C), the most enriched biological processes are response to stimulus (222 interactions), locomotion and taxis (81 interactions), and cell, ciliar, or flagellar motility (33 interactions, Fig 3D). Comparison of the Pearson correlation coefficient (PCC) between the expression profiles of TFs and their targets to random pairs shows the first to be significantly highly correlated (Kolmogorov–Smirnov test $P < 10^{-10}$ and Mann–Whitney test $P < 10^{-10}$; Supplementary Fig S6) and with similar profiles for both experimentally validated and computationally inferred interactions, which reinforces the likelihood that these putative interactions are indeed present in the respective experimental conditions.

### Expression Balance Analysis

Training a regression model on *Eco*MAC was found to be highly predictive of positive (0.74 AUC) and negative (0.91 AUC) interactions for arrays where TFs and genes were significantly correlated (interactions with $PCC > 0.75$, Fig 3C). The EBA model was used to predict genome-wide gene expression values under genetic and environmental perturbations in *Eco*MAC (Supplementary Methods, section 4.4). We analyzed the predictive power of EBA on the entire gene expression profile or a subset of it, resulting in two evaluation classes (global and local, respectively). For local evaluation, all genes with a distance of two links or less from the perturbed gene were considered. After parameter training (Supplementary Fig S12), the EBA model was significantly more accurate in predicting global expression profiles when compared to the null-model (Fig 4A and B). Specifically, the 50 and 64% of well-predicted arrays for genetic and environmental perturbations, respectively, outperformed the *PCC* average of all predictions (437 and 55 arrays evaluated, respectively; Fig 4A, solid area; Fig 4B, blue points), whereas the null-model is shown in (Fig 4A, hatched area; Fig 4B, red points). We also assessed the effect of genetic and environmental constraints in the EBA model by comparing its performance to EBA predictions when no or random constraints are imposed. Although the performance in both these cases is closer to that of the (constraint-driven) EBA model, the latter results in better predictions (measured by the number of arrays above the average PCC threshold) as shown in Fig 4A (bottom panel). Furthermore, the EBA method was found to

be robust to parameter perturbations (Supplementary Fig S13). Similar results were obtained when computationally inferred interactions were included in the analysis (Supplementary Fig S14), and individual classes of genetic perturbations were taken into account (Supplementary Fig S15).

We studied the performance of EBA by training random sub-sets of transcriptional interactions (Supplementary Fig S16A and B). As expected, the EBA local performance decreased significantly when the TRN was constructed by using random interactions between TFs and genes. Moreover, when interactions were excluded from the TRN, an exponential decrease in performance on local profiles was observed that is consistent with the scale-free nature of the TRN network. A fivefold cross-validation argues that EBA is robust in the size and nature of the training set (Supplementary Fig S16C and D).

### Integrating transcription, fluxes, and metabolic models

Next, we used the *E. coli* metabolic model iJO1336 together with Flux Variability Analysis (FVA) (Mahadevan & Schilling, 2003; Gudmundsson & Thiele, 2010) and Flux Balance Analysis (FBA) (Orth *et al*, 2011) to calculate the reaction fluxes and their bounds. In order to test the metabolic model under various environmental conditions, we simulated 100 random environments where cells grew in minimal media and a growth-affecting parameter in abundance or limitation (carbon sources, nitrogen, supplemental amino acids, or metals). In all cases, the model provides a quantitative measure of the variations in growth rate for the different environmental perturbations (Supplementary Fig S17 and Supplementary Dataset S6). TF and enzyme knockouts were found to be phenotypically more diverse than over-expressions, as shown in analysis of the metabolic benefit under single (Supplementary Fig S18) and multiple (Supplementary Fig S19) genetic perturbations. We then used TRAME to integrate metabolic and transcriptional regulatory networks by modifying the metabolic flux bounds (Supplementary Methods, section 5).

### Phenotypic predictions in an integrated model

To integrate all the models described above, we used a cost-benefit scheme across the various layers to determine the genome-scale gene expression profile, metabolic flux distribution, and growth rate. The cost-benefit model outperformed growth predictions of models that contained only benefit or cost-limited functions, with PCC between predicted and measured phenotypes at 0.76 ($P < 10^{-3}$) for benefit-only model predictions versus 0.84 ($P < 10^{-4}$) in our model (Supplementary Fig S21). Interestingly, when inferred interactions were added in the analysis, more arrays were well predicted, leading to a higher PCC between predicted and measured growth rates than when EBA was restricted only to experimental interactions ($PCC > 0.53$, $P < 2 \cdot 10^{-4}$; Supplementary Fig S21C). Fig 4C shows high correlations between predicted and measured growth rates for different categories of arrays in *Eco*MAC. The model accurately predicted growth in all cases with *PCC* ranging from 0.8 (genetic perturbations; $P < 10^{-10}$) to 0.99 (gene knockouts; $P < 10^{-10}$).

Next, we assessed the predictive power of this work in comparison with three recent M-models (Beg *et al*, 2007; Orth *et al*, 2011; Adadi *et al*, 2012) and a ME-model (O'Brien *et al*, 2013) for *E. coli*,

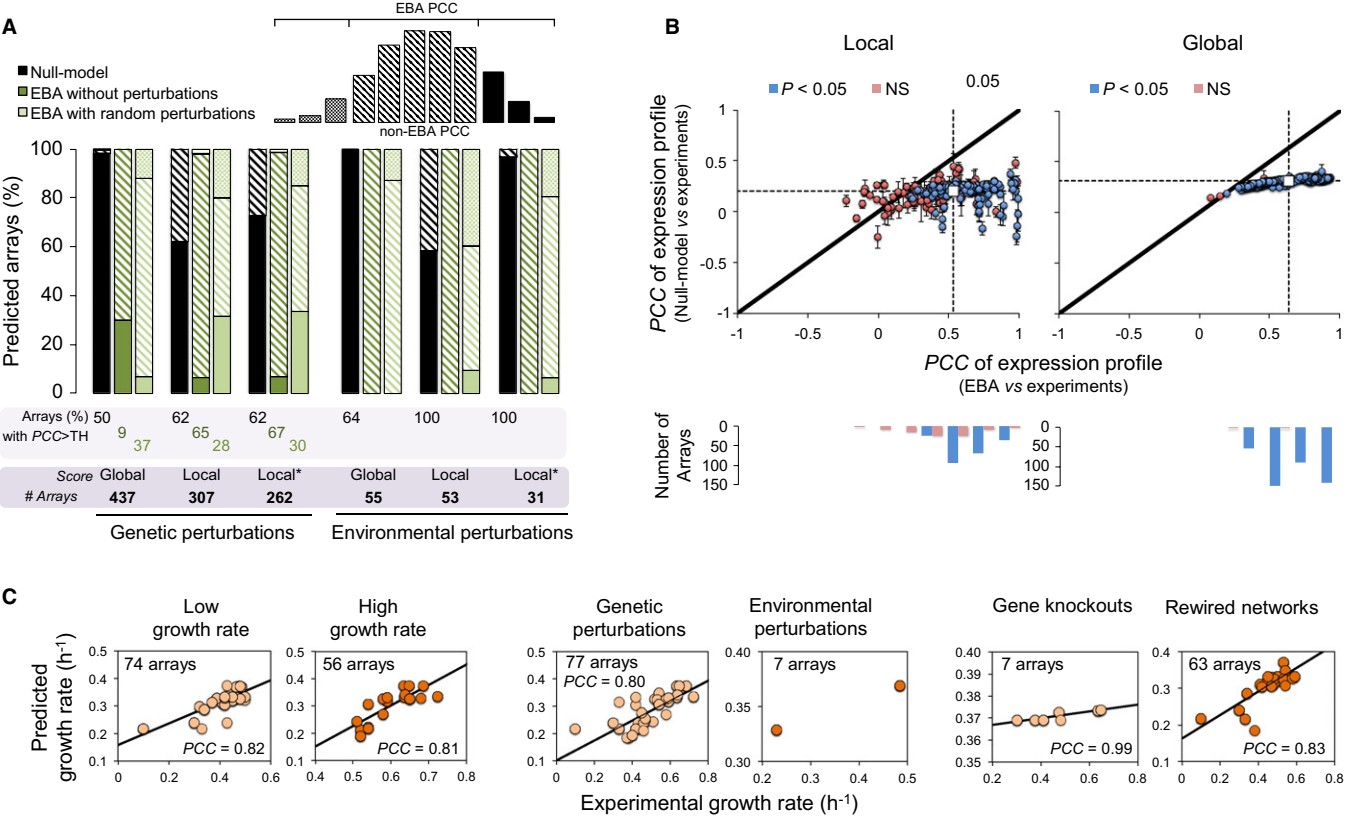

**Figure 4.  Quantitative assessment of Expression Balance Analysis.**

A   Percentage of arrays where EBA achieved a higher (solid bars), equal (non-solid bars with lines), or lower (non-solid bars with pattern) PCC between the predicted and the measured expression profile, when compared to a randomized model. Three randomized models were considered: randomized *Eco*MAC expression profiles (null-model, black bars), expression profiles that are derived by EBA that does not encode the environmental or genetic constraints of the expression profile under investigation (dark green) and EBA-derived expression profiles with random genetic and/or environmental constraints (light green). Both genetic (left panel) and environmental (right panel) perturbations were considered. The percentage of the arrays predicted by EBA with PCC higher than a threshold PCC (noted as TH, calculated as the average over all predicted arrays) is denoted in the bottom panel. Bottom panel contains the number of well-predicted arrays with PCC higher than the average. The comparison is performed for all well-predicted arrays (global), those within a distance of two links (local), and those local arrays with a PCC that is statistically significant (*P*-value < 0.05; local*).

B   Predictive power of EBA for all genetic perturbations by using the null-model (black bars in panel A). Blue and red points show arrays in which the *PCC* between the measured and predicted (EBA) expression profile is significantly higher (*P* < 0.05) or lower (NS, non-significant), respectively, than the *PCC* between the measured expression profile and the null-model (i.e., random profiles from *Eco*MAC).

C   Phenotype predictions for arrays in *Eco*Phe compare different categories of perturbations (low *vs* high growth rate measured; genetic *vs* environmental perturbations; gene knockouts *vs* rewired networks) by using the integrative genome-scale model of *Escherichia coli* in which EBA with experimental and inferred interactions predicted gene expression profiles. PCC corresponds to the correlation between the predicted and experimentally measured growth for each category.

as well as to the first whole-cell model for *M. genitalium* (Karr *et al*, 2012). We used our integrative model to predict growth rates in 14 different batch cultures that can be captured by the model and gene essentiality of all the 4,189 *E. coli* genes considered in our model (Supplementary Dataset S11). Interestingly, the correlation between measured and predicted growth rates by using our model (PCC = 0.60, *P*-value = 0.02) was higher and statistically more significant than for two M-models (PCC = 0.20, *P*-value = 0.49 for the iJO1366 yield model presented in (Orth *et al*, 2011), PCC = 0.36, *P*-value = 0.20 for the FBAwMC model presented (Beg *et al*, 2007)), and ME (PCC = 0.50, *P*-value = 0.07 for the ME-model in (O'Brien *et al*, 2013)). Similarly, *in silico* prediction of gene essentiality in glucose M9 minimal medium results in an accuracy of 91.1% (Supplementary Dataset S11, "Gene Essentiality"). This accuracy is on par with previous approaches using the metabolic reaction

network alone (accuracy = 91.2% reported in Orth *et al*, 2011) and the ME-model (accuracy = 88.8% reported in O'Brien *et al*, 2013), as well as the reported accuracy of 79% on the whole-cell model of *M. genitalium* (Karr *et al*, 2012).

## Model enrichment through targeted experimentation

We explored the landscape of biological processes that could be affected by implementing all genetic perturbations contained in *Eco*MAC. From the 1,361 GO terms associated with biological processes in *E. coli*, we included GO terms belonging to the first five levels of the GO-hierarchy, resulting in a set of 686 GO terms, covering 3,319 *E. coli* genes (80% of total; Supplementary Fig S23A and B). Only 23% of these GO terms are affected by the genetic perturbations present in *Eco*MAC given two coverage constraints (Fig 5A;

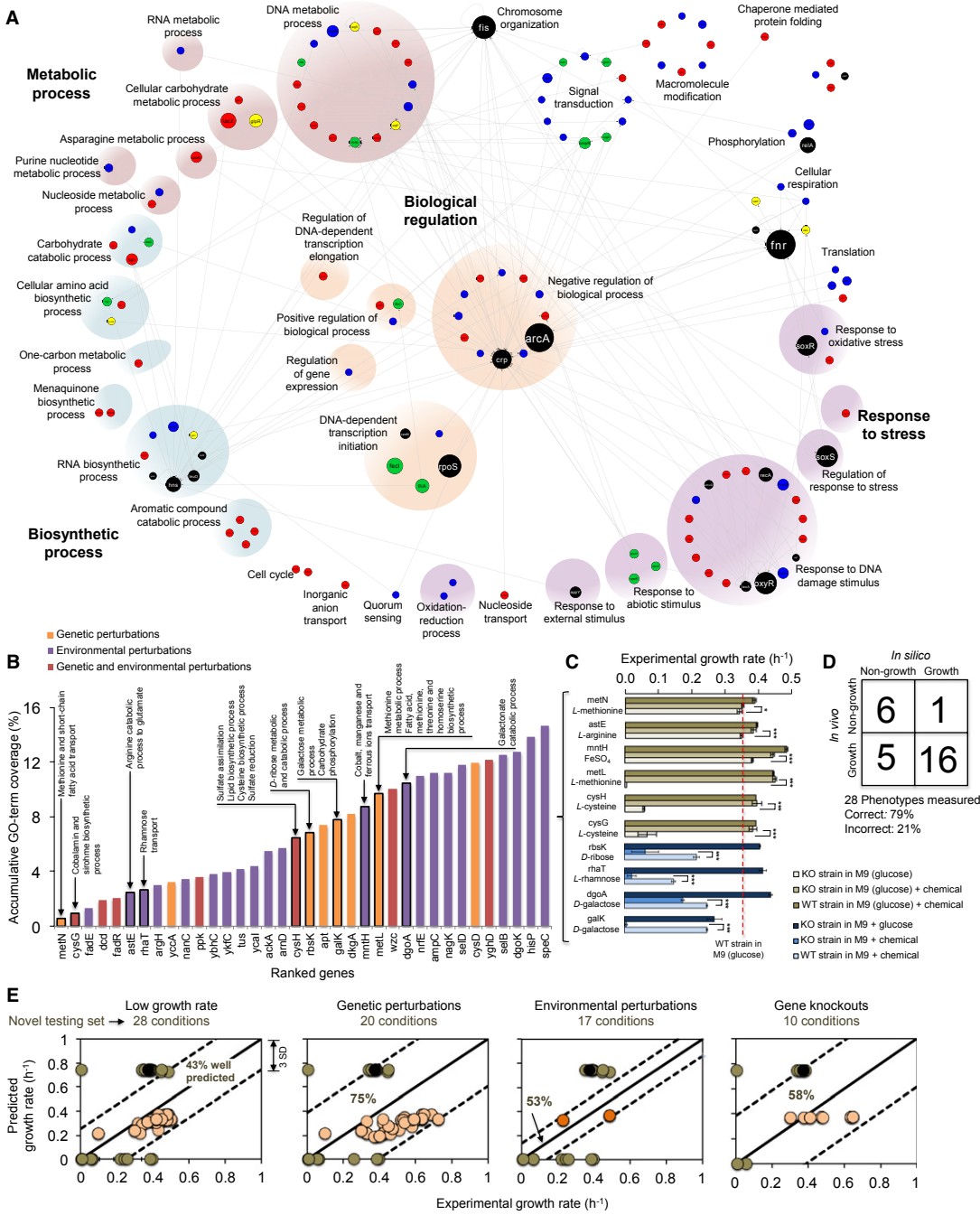

**Figure 5. Model validation.**

A   Perturbed genes in arrays of *Eco*MAC grouped by GO terms altered. The 172 links are the transcriptional interactions identified in the experimental TRN. Note that the 36 low-level GO terms are plotted from the 160 GO terms found to be enriched.

B   Accumulative GO term coverage under the list of the 35 top gene perturbations, according to gene expression variability under genetic (orange), environmental (purple), or gene–environment combinatorial (red) perturbations. The 10 gene knockouts that were experimentally measured and their corresponding under-represented GO terms are highlighted.

C   Growth rates of the 10 gene knockouts in environments with and without supplementation (* $P < 10^{-2}$, ** $P < 10^{-3}$, *** $P < 10^{-10}$). The dashed red line depicts the growth rate for the WT strain in M9 salt, 0.3% glucose media without any other supplements.

D   Comparison of the 28 predicted and observed phenotypes.

E   Phenotypic predictions for the 28 newly measured phenotypes, grouped into categories based on the perturbation type (low growth rate; genetic and environmental perturbations; and gene knockouts). Each panel shows the percentage of accurately predicted conditions. Both validated and inferred interactions were used for training the integrative model (Supplementary Methods, section 7.3 and 7.4). Orange (pink) and green dots depict arrays predicted from Fig 4C and newly measured phenotypes, respectively. Black dots are measurements for the WT strain in non-supplemented M9 media.

Source data are available online for this figure.

Supplementary Methods, section 7.1), a remarkably low number that signifies the limitations of the training set for capturing biological processes by any model. In order to determine the minimal set of gene knockouts that maximizes the GO term coverage and expected gene expression variability, we devised a greedy algorithm (Supplementary Methods, section 7.2; Supplementary Fig S23C and D) that produced a ranked list of gene knockouts that maximize the likelihood for model enrichment by performing expression profiling and growth measurements. Including the top 35 candidate genes improves the number of affected GO terms by a staggering 14.6%, which is in strong contrast to the $3.3 \pm 1.1\%$ that is expected by a random assignment of 35 genes that are currently not present in the dataset (Fig 5B).

To further test the ability of the genome-scale model to predict growth rates under adverse conditions, we isolated 10 single-gene knockouts from these 35 genes that are part of under-represented GO biological processes in *Eco*MAC and for which we had no growth or expression information (Fig 5C, Supplementary Dataset S7). We then computationally predicted and experimentally measured the growth rates of these knockout strains in minimal and supplemented media (28 different combinatorial phenotypes explored; Fig 5C; Supplementary Methods 7.4). In 9 out of the 10 cases, supplementation of the media with the necessary compounds led to significant increase ($P < 10^{-10}$, Z-test) in growth (Supplementary Fig S24). Interestingly, the model captures growth-related defects with 79% accuracy ($P < 10^{-4}$ Fisher's exact test; Fig 5D) across the 28 phenotypes related to strains with gene knockouts in transport genes (methionine and short-chain fatty acids transport, *metN*; rhamnose transport, *rhaT*; cobalt, manganese, and ferrous ions transport, *mntH*), biosynthesis-related genes (cobalamin and siroheme biosynthesis, *cysG*; cysteine biosynthesis, *cysH*; fatty acid, methionine, threonine, and homoserine biosynthesis, *metL*), and metabolic processes (arginine catabolism, c*astE*; D-ribose catabolic process, *rbsK*; galactonate catabolism, *dgoA*; galactose metabolic process and carbohydrate phosphorylation, *galK*). In addition, the model predictions were within the confidence intervals for 75%, 53%, and 58% of the phenotypes related to genetic perturbations, environmental perturbations, and gene knockouts, respectively, despite the fact that these knockouts were not part of the *Eco*MAC dataset and with GO terms that are not represented in the compendium (Fig 5E). Enrichment analysis of the differential expression predicted by the model shows 149 genes were significantly altered ($P < 10^{-3}$ Z-test) and are implicated in signal transduction systems and TRN (response to external stimulus), and metabolism (primary metabolic process, carbohydrate and amino acid metabolic and catabolic process) of *E. coli* (Supplementary Dataset S8). A detailed analysis of the pathways implicated and justification for the observed growth can be found in Supplementary Methods (sections 7.3 and 7.4).

## Discussion

One of the most striking realizations that came to light after constructing the various *E. coli* compendia was the paucity of our knowledge even for the most-studied bacterial organism: when accounting for all gene knockouts, rewirings, or over-expressions, we have data for 141 genes that cover 23% of GO terms, a surprisingly low percentage of coverage. Given that these experiments

have been performed in different strains and experimental conditions, there is a clear and present need for the creation of comprehensive datasets that aim at the construction of more informative models. The idea of targeted experimentation for model enrichment departs from the classical view of experiments as an answer to hypothesis testing and subsequent model training on seemingly disparate datasets. Rather, to maximize the impact on the model's predictive ability, the experimental focus and type can be selected based on current model deficiencies by using the proper heuristics, such as the maximal increase in GO term coverage and gene expression variability that we used in this work.

The *Eco*MAC compendium increased considerably both the sensitivity and specificity of the known inference methods. Indeed, training only three inference methods on *Eco*MAC yielded results that were significantly better than those obtained recently by using 35 methods (Marbach *et al*, 2012), but on a smaller compendium. A limitation here is the severe bias to negative samples in the ground truth: while experimentally confirmed interactions are well documented and categorized, there is no such set for true negatives, that is interactions that were experimentally tested and found non-existent. As such, all inference assessments consider all TF–gene combinations that are not denoted as confirmed, to be negative, hence introducing an artificially high False Positive Rate.

Integration of signal transduction, gene expression, and metabolic levels under one overarching framework led to a significantly more predictive model that can capture environmental and genetic perturbations beyond what was possible before (Monk *et al*, 2013; O'Brien *et al*, 2013). There are several extensions over previous attempts that have made this possible. Having constructed a phenomenological model that focuses more on the statistical associations among the various components and less on the biophysical mechanisms of each individual sub-component, we reduced the parameter space compared to other biophysical models (Segre *et al*, 2002; Beg *et al*, 2007; Lee *et al*, 2008; Adadi *et al*, 2012; Karr *et al*, 2012; Lerman *et al*, 2012; Thiele *et al*, 2012), a step that reduces the amount of over-fitting given the limited availability of experimental data. Instead of using summary statistics to provide kinetic and flow bounds to macromolecular synthesis machinery reactions (Lerman *et al*, 2012; Thiele *et al*, 2012), we here rely on a methodological large-scale analysis of gene expression datasets to capture the dependencies and predictive associations among gene products. This allows the model to generate predictions beyond the subset of gene products related to biosynthesis that have been explicitly modeled and reported in the past. Computationally, the Expression Balance Analysis (EBA) technique that we developed is similar to previous work (Covert *et al*, 2004; Lerman *et al*, 2012) in the sense that it employs constrained optimization, although the actual constraints and objective functions are different, aiming at training the model parameters so that they maximize the likelihood of the data, in a realistic fashion. An important contribution of this work is the creation of a signal transduction network (*Eco*ST) and its integration to the transcriptional and metabolic network through constraint modeling, which enables our model to capture environmental perturbations related to carbon, nitrogen and phosphate sources, oxygen, acids, metals, and other medium supplements. Expanding the fitness function to calculate a relative cost for altered gene expression with respect to wild type allowed our model to predict a wide range of genetic perturbations that includes

transcription factor rewirings to the set of knockout and over-expression perturbations that past models have focused on.

Aside from its merit as a hypothesis-testing tool for systems biology, integration of this work with synthetic circuit and genome redesign platforms (Huynh *et al*, 2013) is a stepping stone toward unifying, model-driven designs that transcend multiple layers of biological function. Further refinement and extension of the supporting compendia will inarguably provide an important knowledgebase for integrative models that exploit associations between heterogeneous genotypic and phenotypic characteristics. While innovative, the current model can be extended to use a mixture of statistical learning techniques that capture different aspects of the data structure. This, together with iterative cycles of training, targeted experimentation and refinement is poised to have a transformative potential on our ability to accurately predict cellular states and generalize in new environments.

# Materials and Methods

## A gene expression, signal transduction, and phenomics compendium for *Escherichia coli*

By integration of microarray data from GEO, ASAP database, Array-Express, and individual investigators, we have constructed a gene expression compendium of 4,189 genes over 2,198 arrays that were collected from 127 scientific articles (Supplementary Methods, section 1.1 and 1.2; Supplementary Fig S1). A total of 328 transcription factors (TFs) and 1,357 enzymes were identified by using RegulonDB. From the 2,198 arrays, 90 were considered as "wild-type" conditions (MG1655 strain, aerobic growth in M9/LB media with 0.3% glucose) and 332 arrays that had experimental settings that deviate considerably from these conditions were classified as "environmental perturbations". Another 718 arrays correspond to "genetic perturbation" experiments, where a knockout, over-expression, or gene rewiring took place (Isalan *et al*, 2008). The resulting *E. coli* Microarray Affymetrix Compendium (*Eco*MAC) includes data from 31 strains and over 15 different media with high gene expression diversity (Supplementary Figs S2 and S3). *Eco*MAC is supplemented by *Eco*Phe (Supplementary Methods, section 1.4), a phenomics compendium that has bacterial growth information for 616 of the arrays in *Eco*MAC.

To identify signal transduction pathways that are responsible for cellular responses to environmental stimuli, we curated the EcoCyc/RegulonDB knowledgebase and then curated relevant literature to identify 151 instances of signal transduction systems (STSs) where the expression level of one or more TFs is regulated by the presence of effector molecules. In the resulting database, *Eco*ST, 71 of these TF-effector interactions fall under one of the following four types of auto-regulation: (a) Type I (28 instances): the TF represses its own expression in the absence of an inducer, while derepression occurs at its presence (e.g., *lldR* and *L*-lactate; Supplementary Figs S7B and S8A), (b) Type II (11 instances): the TF-effector complex regulates its own expression in the presence of the effector (e.g., *fur* and iron; Supplementary Figs S7C and S8B), (c) Type III (4 instances): two component systems where a histidine kinase sensor is auto-phosphorylated in the presence of an effector and transfers the phosphate to a TF that can in turn positively (3 instances) or

negatively (1 instance) regulate its own expression (e.g., *dpiA* and citrates Supplementary Fig S7D), and (d) Type IV (28 instances) where TF gene expression is altered in the presence of the effector but the corresponding mechanism is not known (e.g., *fhlA* and formate; Supplementary Fig S7E). The rest of the 80 signal-mediated regulatory interactions were described in literature, but they did not show a significant change in gene expression levels in presence of the effectors. Supplementary Dataset S5 contains all the signal transduction systems that we considered.

## Gene regulatory network reconstruction

We compiled a list of 3,704 regulatory interactions from RegulonDB v8.1, 115 of which were auto-regulatory interactions (3.1%) (Supplementary Dataset S3). Positive interactions are slightly more represented than negative interactions (1,807 versus 1,664), with 233 interactions being dual in nature. We also created three sets of data based on the confidence level of the interactions (Supplementary Methods, section 2.1): a first set with 566 "confirmed" evidence interactions (existence of two or more types of strong experimental evidence), a second set that includes all 566 confirmed and another 2,517 "strong" evidence interactions (existence of only one type of strong evidence) for a total of 3,083 interactions, and the third set includes all 3,704 interactions, with 711 of them based only on "weak" evidence (Fig 3C). For evaluation, we used three golden standards. First, we used the golden standard used in Marbach *et al*, 2012, which includes interactions with strong evidence from RegulonDB v6.8. The other two testing sets consist of the interactions that are labeled as strong (one type of strong evidence, 3,083 interactions) and confirmed (two types of strong evidence, 556 interactions) based on RegulonDB v8.1, respectively (Fig 3C).

We evaluated five top-ranked regulatory interaction inference methods, and we selected three (GENIE3 (Huynh-Thu *et al*, 2010), TIGRESS (Haury *et al*, 2012) and Inferelator (Greenfield *et al*, 2010)) based on their performance to integrate as a meta-classifier and train with *Eco*MAC. By using the same evaluation criteria with the DREAM5 network inference challenge, we compared the performance of the meta-classifier trained on *Eco*MAC to that trained on the 805-array dataset used in (Marbach *et al*, 2012). Supplementary Fig S5 depicts the ROC curves and AUC values for the meta-classifiers and individual methods for both datasets and for three different golden standards. The resulting consensus network of the first 500 inferred interactions achieves a precision of 0.45, with 381 (76.2%) of them overlapping with previous predictions.

## Cellular sub-models

### Signal transduction model

To model the effect of signal transduction systems (STSs) on gene expression, we considered the cases where the effector's presence alters TF concentration or its structural conformation and functionality. In the case where the effector has a direct impact to the TF's concentration, we defined a linear constraint to describe the TF expression $y_{\mathrm{TF}}$ as a function of changes in effector concentrations $\triangle n_{\mathrm{E}}$:

$$y_{\mathrm{TF}} = y_{\mathrm{TF}}^{\mathrm{wt}} + \Omega(C_{\mathrm{TF}}^{\mathrm{max}} - C_{\mathrm{TF}}^{\mathrm{min}})\chi_{\mathrm{TF}}^{E}\frac{\Delta n_E}{\Delta n_E^{\mathrm{max}}},$$

where $y_{TF}^{wt}$, $C_{TF}^{min}$ and $C_{TF}^{max}$ are the wild-type, minimum and maximum expression values of the TF gene obtained from *Eco*MAC. $\Delta n_E$ is the difference in the effector concentration between the predicted and reference (WT) levels, and $\Delta n_E^{max}$ is an empirical parameter. The parameter $\chi_{TF}^E$ is positive (negative) depending on whether the presence of the effector increases (decreases) the TF concentration, and parameter $\Omega$ was used to fine-tune the STSs (Supplementary Methods, section 3.2). In the second case, we modeled the change in the TF activity by introducing a binary variable $\tau_{TF}^g$ that was zero or one depending whether the TF was still functional after the binding event.

### Transcriptional model and EBA

We model the mRNA dynamics of all genes in the compendium as a function of the TF concentration by linear ordinary differential equations (ODEs). Then, we developed a novel method called "*Expression Balance Analysis*" (EBA) to predict the global gene expression profile of *E. coli* under genetic modifications and environmental changes (Supplementary Methods, section 4). EBA formulates an optimization problem to find the gene expression profile subject to four sets of constraints (phenomenological, capacity, environmental, and genetic constraints). Specifically, we used a fitness function, $E$, that minimizes the gene expression errors of the 328 TFs ($\varepsilon_{TF}$):

Minimize:

$$E = \frac{1}{2}\begin{bmatrix}\bar{y}_{TF} & \bar{\varepsilon}_{TF}\end{bmatrix}\bar{\bar{H}}\begin{bmatrix}\bar{y}_{TF} \\ \bar{\varepsilon}_{TF}\end{bmatrix} + \bar{f}\begin{bmatrix}\bar{y}_{TF} \\ \bar{\varepsilon}_{TF}\end{bmatrix}$$

subject to:

$$\bar{\bar{\Sigma}}\begin{bmatrix}\bar{y}_{TF} \\ \bar{\varepsilon}_{TF}\end{bmatrix} = \bar{\alpha}, \text{ phenomenological constraints;}$$

$$\begin{aligned}\bar{y}_{TF} &\geq \bar{C}_{TF}^{min} \\ \bar{y}_{TF} &\leq \bar{C}_{TF}^{max}\end{aligned}, \text{ capacity constraints;}$$

$$\bar{y}_{TF} = F_G\left(\bar{C}_{min}, \bar{C}_{max}, \bar{\alpha}, \bar{\beta}, \bar{y}_{TF}\right), \text{ genetic constraints;}$$

$$\bar{y}_{TF} = F_E^{(1)}\left(\bar{C}^{min}, \bar{C}^{max}, \bar{y}_{TF}^{wt}, \chi_{TF}^E, \Delta n_E^{max}, \Delta n_E\right)$$
$$\bar{y}_g = F_E^{(2)}\left(\bar{\alpha}, \bar{\beta}, \bar{y}_{TF} \cdot \bar{\tau}_{TF}\right), \text{ environmental constraints;}$$

where $\Sigma = \begin{bmatrix}\text{Id} - \bar{\bar{\beta}} & \text{Id}\end{bmatrix}$, the hessian matrix $H = \begin{bmatrix}\bar{\bar{0}} & \bar{\bar{0}} \\ \bar{\bar{0}} & \bar{I}\end{bmatrix}$, $\bar{f} = \bar{0}$, $\bar{\alpha}$ is a vector of the basal transcription coefficients, and $\bar{\beta}$ is a matrix with elements $\beta_{ij}$ that represent the effect of the $j^{th}$ TF to the $i^{th}$ gene. The maximum ($\bar{C}_{max}$) and minimum ($\bar{C}_{min}$) values of gene expression for each gene were obtained from *Eco*MAC.

### Metabolic model and Transcription-based Flux Enrichment

We created a transcription-based metabolic flux enrichment (TRAME) method to integrate metabolic and transcriptional regulatory networks modifying the $V_{min}$ and $V_{max}$ calculated from Flux Variability Analysis (FVA) for each metabolic flux in the *E. coli* metabolic model iJO1336 (Orth *et al*, 2011). This approach determines the new values of the flux bounds $V_{min}$ and $V_{max}$ for a given enzyme, *e*, as a function of the expression (*P*-function) relative to the WT enzyme expression (Supplementary Methods, section 5), $PV_{min} \leq v \leq PV_{max}$, where $P_e = \left(\frac{y_e}{y_e^{wt}}\right)^n$, and $n$ is a parameter that

allows us to factor in the variability observed on the wild-type arrays regarding the expression of that specific enzyme, $y_e^{WT}$.

### Model integration

Integration of the various cellular and environmental components to phenotypic changes was performed through a cost-benefit model. As such, we compute the growth burden due to the production and maintenance of all proteins (cost), as well as the growth advantage due to the energy uptake of the metabolic pathways in each environment (benefit). Figure 2 and Supplementary Fig S20 depict the information flow among the distinct sub-components in our framework. In this cost-benefit model, the genetic cost is defined as the relative reduction in growth rate ($\mu$) due to the production of essential proteins. We used the EBA method to predict gene expression profiles ($\bar{y}_g$) under environmental and genetic perturbations. To measure the *cost c*, we computed the deviation between the WT ($\bar{y}_g^{WT}$) and predicted ($\bar{y}_g$) gene expression profiles:

$$c = \frac{1}{N_G}\sum_g \left|\frac{\bar{y}_g - y_g^{WT}}{y_g^{WT}}\right|$$

where $N_G$ is the number of genes in *E. coli* genome. Similarly, to compute the *metabolic benefit B*, we used the metabolic sub-model (Supplementary Methods, section 6.1). As such, the fitness function that represents the growth rate $\bar{\mu}$ is given by the difference between the benefit and the cost:

$$\bar{\mu} = B - c$$

Environmental perturbations can modify gene expression through the signal transduction sub-model according to the change of effector concentrations ($\Delta \bar{n}_E$). Similarly, genetic perturbations alter the basal and regulatory coefficients ($\bar{\alpha}$ $\bar{\beta}$) of the respective genes in the transcriptional model. Both environmental and genetic perturbations can directly modify the metabolic fluxes ($\bar{V}_{min}$ $\bar{V}_{max}$).

### Experimental model validation

We identified 10 single-gene knockouts from under-represented GO terms in *Eco*MAC, and we used the *E. coli* model to predict their growth in various environments. Experimental measurements for those single-knockout strains (Keio collection) determined their growth in minimal M9 media under various conditions (with/without supplement carbon sources related to the specific knockout deficiency, and with/without 0.3% glucose; Supplementary Methods, section 7.3). We compared computational predictions to observed growth rates under those 28 phenotypes measured. In addition, we compared predicted and observed growth rates for all the environmental and genetic perturbations included in *Eco*Phe (Supplementary Methods, section 6.2).

**Supplementary information** for this article is available online: http://msb.embopress.org

### Acknowledgements

We thank Arun Durvasula for his help with the curation of the various databases, James Costello for his help with the microarray compendium and

members of the Tagkopoulos laboratory for helpful discussions. We would like to thank Mark Isalan for providing rewiring microarray data and comments on the manuscript. This work was supported by the NSF CAREER award CCF-1254205, OCI-0941360, MCB-1244626, and DOD Army Research Office award W911NF-12-1-023 to IT.

## Author contributions

JC and IT conceived the project. JC, RE, and JL constructed the compendia. JC wrote the code, performed all simulations and analysis. NR performed all wetlab experiments and their analysis. AT performed network inference and enrichment analysis. JC, NR, AT, and IT wrote the paper. IT supervised all aspects of the project.

## Conflict of interest

The authors declare that they have no conflict of interest.

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
