## [Review Process File · Molecular Systems Biology]

An integrative, multi-scale, genome-wide model reveals the phenotypic landscape of *Escherichia coli*

Javier Carrera, Miss Raissa Estrela, Jing Luo, Navneet Rai, Athanasios Tsoukalas, Ilias Tagkopoulos

Corresponding author: Ilias Tagkopoulos, University of California Davis Genome Center

Review timeline:

Submission date:	08 January 2014
Revision received:	24 February 2014
Revision received:	06 April 2014
Editorial Decision:	24 April 2014
Revision received:	02 May 2014
Accepted:	13 May 2014

Editor: Thomas Lemberger

Transaction Report:

1st Revision - authors' response

24 February 2014

Thank you again for submitting your work to Molecular Systems Biology. We have now heard back from the three referees who agreed to evaluate your manuscript. As you will see from the reports below, the referees find the topic of your study of potential interest. They raise, however, substantial concerns on your work, which, I am afraid to say, preclude its publication in its present form.

Without repeating all the points raised by the reviewers, the major issues refer to the following points:

- As it stands, it remains unclear whether the model was conclusively validated or not. It will thus be crucial to demonstrate much more convincingly the performance of the model and to considerably clarify the validation results and methods. Reviewer #2 has numerous requests in this regard.
- It will be essential to include the datasets, models and code related to your model. This would include the following items:
 - o EcoST, EcoPhe, EcoMAC knowledge bases,
 - o the transcriptional regulatory model
 - o the genome-scale model(s) in machine-readable format
 - o the code for EBA, STS and TRAM as well as for the integration of the different components.

These items can be submitted as 'dataset' files or as part of a single zip archive (in which case a text-only README file should be included at the top level to describe all the files in the archive). Sufficient documentation should be provided so that the models and software can be used and run by the community.

On a more editorial level, we would encourage you to provide the source files (Cytoscape or equivalent?) for the network illustrations (eg Figures 3), so that readers can easily download these

networks from the figure. These files can be submitted as 'source data' files (see also msb.embopress.org/guidelines)

If you feel you can satisfactorily deal with these points and those listed by the referees, you may wish to submit a revised version of your manuscript. Please attach a covering letter giving details of the way in which you have handled each of the points raised by the referees. A revised manuscript will be once again subject to review and you probably understand that we can give you no guarantee at this stage that the eventual outcome will be favorable.

REFeree REPORTS:

Reviewer #1:

This paper describes the construction of a model that attempts to unify several levels of biological organization. The model incorporates gene expression, transcriptional regulation, signal transduction and metabolic information. The authors construct and present a methodology for building integrative, multi-scale, genome-wide models, including the development of two novel methods, for gene expression prediction and for flux bound estimation, respectively. The model organism *Escherichia coli* is used to demonstrate the methodology. The resulting model is validated by comparison of its simulation results to experimental growth measurements for specific single gene knockout strains.

The methodology presented represents a considerable contribution in the field of genome-scale modelling, which is rapidly moving towards multi-scale information integration. The paper is very well written, the methodology is described carefully and effectively, and the overall presentation is well-thought out and well-structured. As the authors point out themselves, efforts such as these are an important and valuable step in the direction of building more effective, predictive and integrative models for biological function.

Major essential revisions:

- The main issue with the paper is that despite a wealth of supplementary data and information, there is no model or code submitted for download, either along with the supplementary data, or in a public database for open access. Especially with a contribution of such ambition, I would consider it essential that both the reviewers and, perhaps more importantly, the community have access to these resources.

Other necessary revisions:

- One of the important contributions of this paper is the development of two novel methods: Expression Balance Analysis (EBA) and Transcription-based Metabolic Flux Enrichment (TRAME). Nevertheless, the authors have opted to keep the description of these methods in the supplementary material. I believe these would be of great interest, especially to computational biologists and deserve to be included, or even should be required to be moved to, the main article. I cannot tell if the problem is one of space (total character count was not provided), but even if this was a space-saving decision, perhaps it could be possible to re-arrange the text in order to fit in the novel methodologies, or the editors could provide an opinion on the possibility of slightly exceeding the limit.

- It is claimed that the model presented has "the capacity... to predict environmental and genetic perturbations beyond any stand-alone model currently available". The authors do offer some comparative results, but it should be made clearer in the main article what existing methodologies or models they are comparing their own model to.

- Figures are not presented or referenced in order in the text. This is true both for the 5 figures included with the main article, but also for the 24 supplementary figures (though it does seem that ordering improves as the paper progresses). This makes following the presentation unnecessarily difficult. Contributing to the confusion is the fact that the supplementary figures are not numbered. While this will not be a problem for the main figures, my guess is that the supplementary material

will remain in its current format even upon publication. Considering there is a very large number of supplementary figures, it would help to number them accordingly and perhaps include the figure legends in the same file as the figures.

- It can similarly become confusing in some cases when the authors reference the supplementary methods. Perhaps it would help if the appropriate section was specified, along with the reference. For example, on page 7, in the Model Integration section, it was really difficult to figure out what "... we used the metabolic sub-model (Suppl. Methods)" refers to.

Minor corrections:

- Page 2, line 2: ...such "a" feat...
- Page 2, line 3: ... the potential... transcends most areas of life science research... What is meant by "transcends"? In what other areas of human activity is the potential transformative that are wider than the limits of life sciences?
- Page 2, line 10: "accurate" modeling?
- Page 2, bottom line: I don't believe it is accurate that all references from 16 to 34 describe methods that targeted the integration of models across layers.
- Page 3, line 16: "subsequently" instead of "then"
- Page 3, line 19: "in term", may have been intended to be "in turn"
- Page 11, line 18: ... we included GO terms "that" belonged / or "belonging"
- Page 13: "results... significantly better than those obtained recently by using thirty five methods..." Reference(s) missing.
- Reference 45 may be missing volume and page information.
- Caption for Figure 2: "three new databases" and "two novel computational methodologies"
- Caption for Figure 5, 3rd line: "terms" instead of "termed"?

Reviewer #2:

Carrera et al. gather a comprehensive knowledge-base concerning the model bacterium *E. coli*, and use it to construct a novel genome-scale model of the bacterium, which incorporates transcription, regulation, and metabolism. The effort put into constructing this knowledge-base is commendable, and the latter will no doubt be a useful resource for the research community. However, the paper's main contribution lies in the construction and the validation of the novel integrative model. Whereas the methods employed in constructing the model are methodologically sound, I do have some serious reservations concerning the way it was validated. In particular, I question the soundness of the validations of the EBA component, and of the integrative model as a whole, at least as far as they are described in the manuscript. I would prefer to see those concerns addressed before endorsing this manuscript for publication in MSB. My recommendation is therefore: major revision.

Major comments:

1. The authors do not address properly recent studies of integrative genome-scale modeling.
 - a. The authors do not discuss the differences between their model and recent modeling efforts by the Covert (Karr et al., Cell 2012) and Palsson (Lerman et al., Nat Comm 2012; O'Brien et al., MSB 2013; Thiele et al., PLOS ONE 2012) labs. Rather, they refer to these recent publications only briefly. Such a discussion should include an overview of the differences between their work and the previous publications both with respect to the scope of the model (which cellular subsystems are modeled) and to the computational approach (how each system is modeled, and how is the integration performed).
 - b. The Palsson lab has recently created an integrative *E. coli* model (ME model, cited above). Comparing the predictive power of this model vis-à-vis the predictions of the authors' model will allow better assessment of the contribution of the current manuscript.
 - c. Incidentally, the authors give a secondary citation to the Covert lab's work rather than referring to the primary Cell publication cited above.

2. Validation of the EBA sub-model:

a. The authors validate EBA against 3 null-model. The first is too naïve in my opinion, and therefore not instructive (and EBA always beats it by a wide margin as can be seen from the plots). When it comes to the other two null models, EBA does not seem to really beat them, and so the author's conclusion in the main text that the EBA model "was significantly more accurate when compared to various null-models" seems a bit of an overstatement. From the Fig 4A it seems, at least to me, that EBA obtains in most arrays an equal PCC to that of the null-model. I think that the authors do here a disservice to themselves by relying only on PCC for evaluation. The Pearson coefficient is simply not sensitive enough for minute variations in vectors of that size.

b. I wonder if the above is the reason that the authors present (at least as far as I understand) Fig 4B only for the first null-model. This fact is somewhat hidden in the caption ("... null model, i.e. random profiles from EcoMAC). In any case, the axes are not explained in the caption, nor are the abbreviations (NS = not significant?).

c. In the main text the authors say "EBA performance decreased dramatically when the TRN was constructed by using random interactions..." - this seems an over-statement judging by supp Figure S16A,B. Particularly with respect to the global PCC bars. See my comment that PCC is simply not sensitive enough above, and this is especially true for the global PCC.

d. The caption to Figure 4A,4B: "where EBA showed higher, equal, or lower PCC..." - I assume that the authors meant "higher, equal, or lower PCC than the null model"?

e. Supp figure S13: what is the criterion for an array to be "well-predicted" (better than the null-model? PCC larger than a certain threshold?). Does the y-axis count the number of "well-predicted arrays"? Why doesn't the caption say so? Same comments for Supp figure S14.

f. Supp figure S15: "Red lines dashed show the average of several percentages of predicted arrays" - I'm sorry, I do not understand what the red lines represent. Please rephrase.

g. Supp figure S16: in panel B the p-values of the global and local bars seem to have been mixed. Also note in the legend of the same panel that "same size than" should be "same size as".

3. Integrative validation #1: high-throughput data

a. Since 10 KO's are far from being sufficient validation for a genome-scale model, validating the integrative model's prediction against high-throughput data is crucial. I would recommend conducting a cross-validation based on arrays in EcoMAC that had growth phenotypes. The authors seem to have done that (Fig 4C, Supp Fig S21), but the text that describes this part is utterly incomprehensible and I cannot assess the soundness of the validation scheme, nor its results.

b. The main text refers to this validation only briefly and the text is all but unintelligible. The supplementary text (section 6.2) does not help either - it says that the authors compute $PCC(\rho)$ - but what is ρ ? And then: "... the subset that includes... arrays in which growth rate was observed well predicted" - the grammar here is impossible...

c. The main text says: "The number of arrays with high PCC ($PCC > 0.85...$) between measured and predicted growth rate". This leads the reader to assume that there is a PCC assigned to each array - If so, how is it computed? On the other hand, the caption to Fig 4C says "PCC between growth rates measured and predicted growth rates for all arrays" - which means that there is one PCC. The latter is corroborated by the figure, in which the PCC of the model's predictions' PCC has no error bar, and by what seems to be the intention of the supp text (but why do I have to guess...?).

d. I assume the latter option is true - in this case, I suggest removing the panels from figure 4C that show the (single) PCC in a bar plot - I find it confusing, and it's enough to show the scatter plot and state the PCC. I don't think that the null model contributes here as well (as detailed above) and so removing the bar plot does not omit any information.

e. The paragraph "Phenotypic predictions in an integrated model" refers to fig 4B where it should refer to fig 4C.

f. Supp Fig S21: Again, I think that a scatter plot that demonstrates the PCC will be better than a bar plot.

g. Supp Fig S21: What are the numbers below the panels? The caption doesn't mention them. The main text is cryptic ("arrays with high PCC") as mentioned above. Are these the optimal subsets that supp text section 6.2 refers to in passing?

h. Supp Fig S21: Are the black bars the null model? The caption doesn't say

i. Supp Fig S21 caption: "EBA was replaced by the expression profiles from EcoMAC" - do the authors mean that EBA was skipped completely and the "correct" expression profile was used instead? Or do they mean "by a random expression profile from EcoMAC"?

j. Supp Fig S21: the meaning of the asterisks is not given in the caption and has to be gleaned from the main text. All in all - the caption of Fig S21 is lacking - please add details, or refer to the caption of Fig 4C where some details are given.

4. Integrative validation #2: targeted experimentation

a. The authors give a detailed description of the function of each of their selected KOs in the supplementary. However, they do not mention each gene's role in their modeling scheme: are all these genes part of the iJO1366 metabolic model? Or are there genes that encode TFs among them? Were the media supplements chosen such that they will introduce changes through the signal transduction sub-model? An overview of these issues is crucial for the reader to assess the coverage of different modeled sub-systems in the group of tested KOs.

b. I spotted many genes in the set that are already covered by the iJO1366 metabolic model. This raises the immediate question of whether iJO1366 alone could have predicted the KO outcomes. An analysis of this issue will allow the reader to assess the contribution of the integrative model over the well-known and widely-used metabolic model.

c. A comparison of the authors' model results and the E. coli ME model cited above would also be instructive.

d. Figure 5E: the color code of the dots is not explained.

e. Figure 5E: In addition, the methodology of assessing the model's predictions quality is either flawed or incomprehensible. As far as I understand the authors consider a prediction successful if it matches the measured growth rate to a level of 3 STDs, which might be too lenient... In particular, this might not be the best choice, seeing that many KOs produce the same predicted growth rate in silico (Fig 5E). In addition I do not see how do 53% of the measurements of the 3rd panel fall within this lenient bound - it seems that there are only 4 dots within the bounds. On the other hand the description of the method in the supp is incomprehensible so I might be missing something - please revise this paragraph ("the model predictions show discrepancies with measured growth rates non-significant in less than 67% of the phenotypes..."). Also in this place: is μ the experimental growth rate?). The authors' cause might be better served if they compared their predictions to previous

results, since the current predictions might not seem impressive judged to themselves, but might turn out to be much better than previous work allowed.

f. The authors do not describe, as far as I can tell, how growth rate was quantified from the growth curves, nor do they say how many replicates were there of each experiment.

Minor comments:

1. I found the flow of the paper quite perplexing at first read. The abstract focuses so much on the integrative model, and yet I was halfway through the text before any integrative result was mentioned. Adding an overview of the manuscript's flow to the introduction, similar to the one hidden in the supplementary, will enhance readability (i.e., the paper opens with a description of the knowledge-base, continues with validation of each component in isolation, and then validates the integrative model as a whole).
2. Adding code that implements EBA, TRAME, and their integration as supplementary material will greatly enhance the usability of the model for the research community.
3. Numerous times, the authors give a p-value, but do not say what is the statistical test. For example, in the subsection "Model enrichment through targeted experimentation": "...with 79% percent accuracy ($p < 10^{-4}$)" - but what is the test? (Fisher exact?). Same subsection: "... led to high significant ($p < 10^{-10}$) increase in growth". (Also note the grammatical error: "high significant increase")
4. There was a mix-up of the references in the supplementary text: the reference list is bulleted and not numbered; sometimes the citations are of the form (Orth et al., 2011) and sometimes they are numbered.
5. The authors use several times the word "predictability" where they seem to mean "predictive power" - e.g., supp text 4.1: "the predictability of the model increases proportionally to the correlation between..."
6. The authors misspell numerous times the word "were" as "where" - e.g., in the Discussion section, 2nd paragraph: "interactions that where experimentally tested". This occurs both in the main text and in the supp text (supp text 5.1: "lower bounds on B where set to...")
7. The proper reference to FBA (both in main text and in the supp) is not Orth et al., MSB 2011. The authors should cite the original Varma & Palsson 1993 papers, and possibly the review in Orth et al., Nat Biotech 2010.
8. Supp text 4.4.1: mixed word order "to predict global the gene expression".
9. Incidentally, the authors could mention when discussing EBA that the approach of minimizing the change to the WT state following a perturbation is widely used in genome-scale modeling - this is the rationale of the MOMA method, Segre et al., PNAS 2002.
10. Supp text 4.4.1: shouldn't it be y_{TF} rather than y_g in the left-hand side of equation 16?
11. Supp text 4.4.2: shouldn't Z_i in equations 17-19 be replaced with 3?
12. Supp text 4.4.2: "we added a second criterion to characterize well-predicted arrays as those in which P was higher than a threshold" - what is P in this case? Do the authors mean the PCC?
13. Supp text 5.3: equation 22 introduces the parameter n, but I could not find where its value is given.

Reviewer #1 (Remarks to the Author):

This paper describes the construction of a model that attempts to unify several levels of biological organization. The model incorporates gene expression, transcriptional regulation, signal transduction and metabolic information. The authors construct and present a methodology for building integrative, multi-scale, genome-wide models, including the development of two novel methods, for gene expression prediction and for flux bound estimation, respectively. The model organism Escherichia coli is used to demonstrate the methodology. The resulting model is validated by comparison of its simulation results to experimental growth measurements for specific single gene knockout strains.

The methodology presented represents a considerable contribution in the field of genome-scale modeling, which is rapidly moving towards multi-scale information integration. The paper is very well written, the methodology is described carefully and effectively, and the overall presentation is well thought out and well-structured. As the authors point out themselves, efforts such as these are an important and valuable step in the direction of building more effective, predictive and integrative models for biological function.

Major essential revisions:

1. The main issue with the paper is that despite a wealth of supplementary data and information, there is no model or code submitted for download, either along with the supplementary data, or in a public database for open access. Especially with a contribution of such ambition, I would consider it essential that both the reviewers and, perhaps more importantly, the community have access to these resources.

Response: We would like to thank the reviewer for sharing our vision for integrated genome-scale modeling. The knowledgebase has been submitted as supplementary files and the file descriptions were referenced in the Suppl. Text. We have extended the description and the metadata of the supplementary files to make it easier for the public to use the compendia that have been generated in this work. As such, these are the following: EcoMAC, Suppl. File 1; EcoPhe, Suppl. File 2; EcoSTS, Suppl. File 5. In addition, we provide the list of experimentally-verified (Suppl. File 3) and EcoMAC-inferred (Suppl. File 4) transcriptional interactions, the metabolic sources and experiments used for environmental prediction (Suppl. File 6), ranked list of genes for targeted experimentation analysis (Suppl. File 7) and the subset of genes/biological processes that were impacted after our experimental measurements were integrated to the dataset (Suppl. File 8), experimental growth measurements (Suppl. File 9).

In addition, we have added the Matlab code for the integrative model, containing both the EBA and TRAME methods (Suppl. File 10) along with a README file on how to execute the code and the Cytoscape files for the networks shown in Figures 3 and 5 of the main manuscript. Finally, in Suppl. File 11, we now include the correlation values and gene significance from our comparisons between the M/ME models and our integrative model (as requested by reviewer 2).

Other necessary revisions:

2. One of the important contributions of this paper is the development of two novel methods: Expression Balance Analysis (EBA) and Transcription-based Metabolic Flux Enrichment (TRAME). Nevertheless, the authors have opted to keep the description of these methods in the supplementary material. I believe these would be of great interest, especially to computational biologists and deserve to be included, or even should be required to be moved to, the main article. I cannot tell if the problem is one of space (total character count was not provided), but even if this was a space-saving decision, perhaps it could be possible to re-arrange the text in order to fit in the novel methodologies, or the editors could provide an opinion on the possibility of slightly exceeding the limit.

Response: We agree that a description of these methods in the main text is desirable and would be of interest. Although previous versions of the manuscript had a brief description of both TRAME and EBA in the main manuscript, the authors decided at the time to remove it from the methods section due to its increased complexity and semantics that warrant a more extended description, as we have it currently in the supplementary material. We were able to add back a succinct description of these two methods in the Materials and Methods of the main paper (under the sub-section “Cellular sub-models”), which is now moved to the end of the manuscript.

3. It is claimed that the model presented has "the capacity... to predict environmental and genetic perturbations beyond any stand-alone model currently available". The authors do offer some comparative results, but it should be made clearer in the main article what existing methodologies or models they are comparing their own model to.

Response: We have added the necessary clarifications in the second and third paragraphs of the Discussion section. More specifically, we reference (i) the methods that have been used previously to infer transcriptional interactions from large-scale gene expression datasets and (ii) the existing methodologies that integrate gene expression, transcriptional regulation, signal transduction and metabolic information in *E. coli*, including the work of Orth et al., MSB, 2011 [1], O’Brien et al., MSB, 2013 [2], Adadi et al., PLoS Comp. Biol., 2012 [3] and Beg et al., PNAS 2007 [4].

4. Figures are not presented or referenced in order in the text. This is true both for the 5 figures included with the main article, but also for the 24 supplementary figures (though it does seem that ordering improves as the paper progresses). This makes following the presentation unnecessarily difficult. Contributing to the confusion is the fact that the supplementary figures are not numbered. While this will not be a problem for the main figures, my guess is that the supplementary material will remain in its current format even upon publication. Considering there is a very large number of supplementary figures, it would help to number them accordingly and perhaps include the figure legends in the same file as the figures.

Response: Both the (5) main figures and the (24) supplementary figures are now referenced in order, in the main manuscript and supplementary material, respectively. Although desirable and pursued wherever possible, the authors believe that it disrupts the flow of where information is presented if we enforce the Suppl. Figures to be indexed in the order they are referenced in the main manuscript, instead to their reference in the Suppl. Material. Given their volume and size, we created a single pdf file for review and we will be happy to work with the editorial team to get all Suppl. Figures in the Suppl. Material file and include the corresponding caption under each figure, after the final acceptance of the manuscript.

5. It can similarly become confusing in some cases when the authors reference the supplementary methods. Perhaps it would help if the appropriate section was specified, along with the reference. For example, on page 7, in the Model Integration section, it was really difficult to figure out what "... we used the metabolic sub-model (Suppl. Methods)" refers to.

Response: We agree and we now have added the number of the section referenced to the supplementary method.

6. Minor corrections:

•Page 2, line 2: ...such "a" feat...

- Page 2, line 3: ... the potential... transcends most areas of life science research... What is meant by "transcends"? In what other areas of human activity is the potential transformative that are wider than the limits of life sciences?
- Page 2, line 10: "accurate" modeling?
- Page 2, bottom line: I don't believe it is accurate that all references from 16 to 34 describe methods that targeted the integration of models across layers.
- Page 3, line 16: "subsequently" instead of "then"
- Page 3, line 19: "in term", may have been intended to be "in turn"
- Page 11, line 18: ... we included GO terms "that" belonged / or "belonging"
- Page 13: "results... significantly better than those obtained recently by using thirty five methods..." Reference(s) missing.
- Reference 45 may be missing volume and page information.
- Caption for Figure 2: "three new databases" and "two novel computational methodologies"
- Caption for Figure 5, 3rd line: "terms" instead of "termed"?

Response: Corrected throughout the text. We would like to thank the reviewer for his helpful and thorough review.

Reviewer #2 (Remarks to the Author):

Carrera et al. gather a comprehensive knowledge-base concerning the model bacterium E. coli, and use it to construct a novel genome-scale model of the bacterium, which incorporates transcription, regulation, and metabolism. The effort put into constructing this knowledge-base is commendable, and the latter will no doubt be a useful resource for the research community. However, the paper's main contribution lies in the construction and the validation of the novel integrative model. Whereas the methods employed in constructing the model are methodologically sound, I do have some serious reservations concerning the way it was validated. In particular, I question the soundness of the validations of the EBA component, and of the integrative model as a whole, at least as far as they are described in the manuscript. I would prefer to see those concerns addressed before endorsing this manuscript for publication in MSB. My recommendation is therefore: major revision.

Major comments:

1. The authors do not address properly recent studies of integrative genome-scale modeling.
 - a. The authors do not discuss the differences between their model and recent modeling efforts by the Covert (Karr et al., Cell 2012) and Palsson (Lerman et al., Nat Comm 2012; O'Brien et al., MSB 2013; Thiele et al., PLOS ONE 2012) labs. Rather, they refer to these recent publications only briefly. Such a discussion should include an overview of the differences between their work and the previous publications both with respect to the scope of the model (which cellular subsystems are modeled) and to the computational approach (how each system is modeled, and how is the integration performed).

Response: We have expanded the discussion section to reflect the differences between this work and previous studies. More specifically, there are several extensions over previous attempts and novel methods that are introduced in this work. Having constructed a phenomenological model that focuses more on the statistical associations among the various components and less on the biophysical mechanisms of each individual sub-component, we reduced the parameter space compared to other biophysical models, a step that reduces the amount of over-fitting given the limited availability of experimental data. Instead of using summary statistics to provide kinetic

and flow bounds to macromolecular synthesis machinery reactions, we here rely on a methodological large-scale analysis of gene expression datasets to capture the dependencies and predictive associations among gene products. This allows the model to generate predictions beyond the subset of gene products related to biosynthesis that have been explicitly modeled and reported in the past. Computationally, the Expression Balance Analysis (EBA) technique that we developed is similar to previous work in the sense that it employs constrained optimization, although the actual constraints and objective function are different, aiming at training the model parameters so that they maximize the likelihood of the data, in a realistic fashion. An important contribution of this work is the creation of a signal transduction network (EcoST) and its integration to the transcriptional and metabolic network through constraint modeling, which enables our model to capture environmental perturbations related to carbon, nitrogen and phosphate sources, oxygen, acids, metals and other medium supplements. Expanding the fitness function to calculate a relative cost for altered gene expression with respect to wild-type, allowed our model to predict a wide range of genetic perturbations that includes transcription factor rewirings to the set of knock-out and over-expression perturbations that past models have focused on.

b. The Palsson lab has recently created an integrative E. coli model (ME model, cited above). Comparing the predictive power of this model vis-à-vis the predictions of the authors' model will allow better assessment of the contribution of the current manuscript.

Response: We have compared the predictive power of our model with respect to two M-models (Beg et al, 2007; Orth et al, 2011) and the mentioned ME-model (O'Brien et al, 2013) in *E. coli*, and the first whole-cell model in *M. genitalium* (Karr et al., 2012). To do that, we used our integrative model to predict (1) growth rates in 14 different batch cultures (as identified from the O'Brien et al, 2013), and (2) gene essentiality of the 4,189 *E. coli* genes considered in our model. We added a new Supplementary File 11 containing all this information.

In the first analysis, we compared our predictions with respect to two M-models and the mentioned ME-model. Interestingly, the correlation between measured and predicted growth rates by using our model ($PCC = 0.60$, p -value = 0.023) were higher and with more statistical significance than M ($PCC = 0.36$, p -value = 0.199, FBAwMC; $PCC = 0.20$, p -value = 0.489, iJO1366 Yield) and ME ($PCC = 0.50$, p -value = 0.069) models (Supplementary File 11, "Parameters Integrative Model" and "Growth Correlations" tabs).

In the second analysis, we predicted the growth rate of the 4,189 gene knockouts included in our model to study gene essentiality in *E. coli*. *In silico* prediction of gene essentiality in glucose M9 minimal media results in an accuracy of 91.14% (Supplementary File 11, "Gene Essentiality"). This accuracy is on par with previous approaches using the metabolic reaction network alone (accuracy = 91.2%, Orth et al, 2011) or the ME model (accuracy = 88.8%, O'Brien et al, 2013). We should also note that gene essentiality evaluated on other smaller organism with the whole-cell model from the Covert group (Karr et al., 2012) provided similar accuracy levels (79%).

c. Incidentally, the authors give a secondary citation to the Covert lab's work rather than referring to the primary Cell publication cited above.

Response: Corrected throughout the text.

2. Validation of the EBA sub-model:

a. The authors validate EBA against 3 null-model. The first is too naïve in my opinion, and therefore not instructive (and EBA always beats it by a wide margin as can be seen from the plots). When it comes to the other two null models, EBA does not seem to really beat them, and so the author's conclusion in the main text that the EBA model "was significantly more accurate when compared to various null-models" seems a bit of an overstatement. From the Fig 4A it seems, at least to me, that EBA obtains in most arrays an equal PCC to that of the null-model. I think that the authors do here a disservice to themselves by relying only on PCC for evaluation. The Pearson coefficient is simply not sensitive enough for minute variations in vectors of that size.

Response: We agree with the reviewer that EBA clearly over-performs against the first null-model, i.e. comparison with the PCC obtained between random EcoMAC arrays and the experimentally measured profile. Although this is not especially surprising given that EBA is based on standard statistical techniques and biologically realistic assumptions, we argue that a comparison with this null-model is necessary to ensure that the observed performance is not an artefact of relatively invariant expression in the EcoMAC compendium or in the training set. Regarding the second and third null-models, a tighter gap between performances is expected, since both of these models also use the EBA method to predict gene expression. The comparison to these two null-models serves the purpose of demonstrating the effect of the genetic and environmental constraints to the EBA methodology, and less so to evaluate EBA itself against a random model, which is the case in the first comparison. Still, EBA results in better predictions (measured by number of arrays above the PCC threshold) than any of the three models, as shown by the bottom panel of Figure 4A. We have revised the corresponding paragraph in the main text (Expression Balance Analysis section) to reflect this point.

We also agree with the reviewer that PCC is not the best metric to compare large, relatively invariant vectors. To avoid inefficiencies when we use the PCC for EBA evaluation (i.e., invariance of PCC for large vectors), we used two types of scores, global and local, that correspond to different subsets of genes. We considered local genes as those located with a distance of two links/hops from the specific gene perturbed in the array. By assessing local scores, we capture high variations in the PCC since the reduced vector size is substantial, due to the scale-free properties of *E. coli*'s regulatory network.

b. I wonder if the above is the reason that the authors present (at least as far as I understand) Fig 4B only for the first null-model. This fact is somewhat hidden in the caption ("... null model, i.e. random profiles from EcoMAC). In any case, the axes are not explained in the caption, nor are the abbreviations (NS = not significant?).

Response: Fig. 4B indeed depicts the comparison of EBA with respect to the random null-model, for the reasons stated above, i.e. it provides a comparison between an EBA and the most basic EBA-method (while the other two null-models still use unconstrained/randomly constrained EBA). We have revised the caption in Figure 4 and clarified this point.

c. In the main text the authors say "EBA performance decreased dramatically when the TRN was constructed by using random interactions..." - this seems an over-statement judging by supp Figure S16A,B. Particularly with respect to the global PCC bars. See my comment that PCC is simply not sensitive enough above, and this is especially true for the global PCC.

Response: Corrected throughout the text. See answer for comment 2a.

d. The caption to Figure 4A,4B: "where EBA showed higher, equal, or lower PCC..." - I assume that the authors meant "higher, equal, or lower PCC than the null model"?

Response: Correct. We have revised the caption for Figure 4 to clarify this point. Corrected throughout the text.

e. Supp figure S13: what is the criterion for an array to be "well-predicted" (better than the null-model? PCC larger than a certain threshold?). Does the y-axis count the number of "well-predicted arrays"? Why doesn't the caption say so? Same comments for Supp figure S14.

Response: The criterion for the Supp. Figure S13 is better than the null-model (Fig. 4A, black bars). In the case of the Supp. Figure S14, the criterion is different depending on the color of the bar: black (null-model), dark green (EBA without perturbations), or slight green (EBA with random perturbations). The term "well-predicted" is defined in sections 4.4.2 and the figure captions have been updated to reflect this.

f. Supp figure S15: "Red lines dashed show the average of several percentages of predicted arrays" - I'm sorry, I do not understand what to the red lines represent. Please rephrase.

Response: The red lines show the percentage average for each case (global, local, local with significantly better PCC). Corrected in the caption.

g. Supp figure S16: in panel B the p-values of the global and local bars seem to have been mixed. Also note in the legend of the same panel that "same size than" should be "same size as".

Response: Figure corrected.

3. Integrative validation #1: high-throughput data

a. Since 10 KOs are far from being sufficient validation for a genome-scale model, validating the integrative model's prediction against high-throughput data is crucial. I would recommend conducting a cross-validation based on arrays in EcoMAC that had growth phenotypes. The authors seem to have done that (Fig 4C, Supp Fig S21), but the text that describes this part is utterly incomprehensible and I cannot assess the soundness of the validation scheme, nor its results.

Response: We implemented a cross-validation based on selecting gene expression profiles from EcoMAC to assess the EBA performance (Suppl. Fig. S16; Suppl. Fig. 21). We have clarified the related sections in the main manuscript (page 6, lines 17-18) and SOM text. Please note that we do not use growth rate data to train our model or any sub-model. We only use growth information to test our predictions focused on the arrays from EcoMAC where the growth rate was annotated (i.e., EcoPhe).

b. The main text refers to this validation only briefly and the text is all but unintelligible. The supplementary text (section 6.2) does not help either - it says that the authors compute $PCC(\rho)$ - but what is ρ ? And then: "... the subset that includes... arrays in which growth rate was observed well predicted" - the grammar here is impossible...

Response: The text refers to the Pearson Correlation Coefficient (PCC) and the text in both main text and supplementary materials has been revised accordingly.

c. The main text says: "The number of arrays with high PCC ($PCC > 0.85...$) between measured and predicted growth rate". This leads the reader to assume that there is a PCC assigned to each array - If so, how is it computed? On the other hand, the caption to Fig 4C says "PCC between growth rates measured and predicted growth rates for all arrays" - which means that there is one PCC. The latter is corroborated by the figure, in which the PCC of the model's predictions' PCC has no error bar, and by what seems to be the intention of the supp text (but why do I have to guess...?).

Response: The text refers to the fraction of arrays that are predicted. The reviewer is correct on his assumption that there is only one PCC (per category) and we have revised both the main text (page 7, lines 4-16) and SOM to reflect this fact.

d. I assume the latter option is true - in this case, I suggest removing the panels from figure 4C that show the (single) PCC in a bar plot - I find it confusing, and it's enough to show the scatter plot and state the PCC. I don't think that the null model contributes here as well (as detailed above) and so removing the bar plot does not omit any information.

Response: We have modified the Figure 4C and Supp. Figure 22 as suggested.

e. The paragraph "Phenotypic predictions in an integrated model" refers to fig 4B where it should refer to fig 4C.

Response: Corrected throughout the text.

f. Supp Fig S21: Again, I think that a scatter plot that demonstrates the PCC will be better than a bar plot.

Response: We have added another sub-panel in the Suppl. Figure S21 showing scatter plots to demonstrate the PCC of the panel C.

g. Supp Fig S21: What are the numbers below the panels? The caption doesn't mention them. The main text is cryptic ("arrays with high PCC") as mentioned above. Are these the optimal subsets that supp text section 6.2 refers to in passing?

Response: The reviewer is correct, these numbers correspond to the size of the optimal subsets of arrays, as mentioned in section 6.2. We have added this information and revised the caption of Figure S21.

h. Supp Fig S21: Are the black bars the null model? The caption doesn't say

Response: Correct, we have revised the caption and relevant text to reflect this.

i. Supp Fig S21 caption: "EBA was replaced by the expression profiles from EcoMAC" - do the authors mean that EBA was skipped completely and the "correct" expression profile was used instead? Or do they mean "by a random expression profile from EcoMAC"?

Response: The first is true. We have revised the Suppl. Fig. S21 caption to reflect this clarification.

j. Supp Fig S21: the meaning of the asterisks is not given in the caption and has to be gleaned from the main text. All in all - the caption of Fig S21 is lacking - please add details, or refer to the caption of Fig 4C where some details are given.

Response: We have revised the caption of Suppl. Fig. 21 to clarify what each point represents.

4. Integrative validation #2: targeted experimentation

a. The authors give a detailed description of the function of each of their selected KOs in the supplementary. However, they do not mention each gene's role in their modeling scheme: are all these genes part of the iJO1366 metabolic model? Or are there genes that encode TFs among them? Were the media supplements chosen such that they will introduce changes through the signal transduction sub-model? An overview of these issues is crucial for the reader to assess the coverage of different modeled sub-systems in the group of tested KOs.

Response: Yes, all genes are part of the metabolic model (now clarified in page 24 of the SOM, lines 21-28). We have provided a detailed description of the function of each gene knockout on the integrative model in the section 7.3 of the Suppl. Methods.

b. I spotted many genes in the set that are already covered by the iJO1366 metabolic model. This raises the immediate question of whether iJO1366 alone could have predicted the KO outcomes. An analysis of this issue will allow the reader to assess the contribution of the integrative model over the well-known and widely-used metabolic model.

Response: Suppl. Figure S21 precisely shows a comparison of the predictions by using the integrative model (i.e., benefit-cost model) with respect to two models accounting only the genetic cost or the metabolic benefit provided by EBA or FBA, respectively. As shown from these results, FBA with iJO1366 alone has a significantly lower predictive ability: its PCC between predicted and measured is 0.76 (p-value < 1e-3) vs. 0.84 (p-value < 1e-4) in our model as shown in Suppl. Fig. 21C

c. A comparison of the authors' model results and the E. coli ME model cited above would also be instructive.

Response: As discussed before, we addressed this point by predicting all single gene knockouts of *E. coli* and comparing the predictive power (accuracy) with respect to M and ME-models. Please see the response to point 1.b for more details.

d. Figure 5E: the color code of the dots is not explained.

Response: Corrected throughout the text.

e. Figure 5E: In addition, the methodology of assessing the model's predictions quality is either flawed or incomprehensible. As far as I understand the authors consider a prediction successful if it matches the measured growth rate to a level of 3 STDs, which might be too lenient... In particular, this might not be the best choice, seeing that many KOs produce the same predicted growth rate in silico (Fig 5E). In addition I do not see how do 53% of the measurements of the 3rd panel fall within this lenient bound - it seems that there are only 4 dots within the bounds. On the other hand the description of the method in the supp is incomprehensible so I might be missing something - please revise this paragraph ("the model predictions show discrepancies with measured growth rates non-significant in less than 67% of the

phenotypes....". Also in this place: is μ the experimental growth rate?). The authors' cause might be better served if they compared their predictions to previous results, since the current predictions might not seem impressive judged to themselves, but might turn out to be much better than previous work allowed.

Response: We have revised the caption of the Fig. 5E and the main text to clarify the predictive power in each sub-panel. Note that the predictive power of the all panels (e.g., 3rd panel – 53%) was calculated by accounting only the new predicted arrays, i.e., brown dots. We have considered a prediction successful if it matches the measured growth rate to a level of 3 STDs that is the same criterion used on the Carr et al. model (5). Please see the comparison to previous work in our responses above.

f. The authors do not describe, as far as I can tell, how growth rate was quantified from the growth curves, nor do they say how many replicates were there of each experiment.

Response: We have clarified this point in the section 7.3 of the Suppl. Methods.

Minor comments:

1. I found the flow of the paper quite perplexing at first read. The abstract focuses so much on the integrative model, and yet I was halfway through the text before any integrative result was mentioned. Adding an overview of the manuscript's flow to the introduction, similar to the one hidden in the supplementary, will enhance readability (i.e., the paper opens with a description of the knowledge-base, continues with validation of each component in isolation, and then validates the integrative model as a whole).

Response: Corrected throughout the text.

2. Adding code that implements EBA, TRAME, and their integration as supplementary material will greatly enhance the usability of the model for the research community.

Response: We provided a new supplementary file containing the code and the knowledge-base.

3. Numerous times, the authors give a p-value, but do not say what is the statistical test. For example, in the subsection "Model enrichment through targeted experimentation": "...with 79% percent accuracy ($p < 10^{-4}$)" - but what is the test? (Fisher exact?). Same subsection: "... led to high significant ($p < 10^{-10}$) increase in growth". (Also note the grammatical error: "high significant increase")

Response: The nature of the statistical test (Fisher exact, z-test, Kolmogorov-Smirnov, Mann-Whitney test) is now referenced throughout the main and supplementary text.

4. There was a mix-up of the references in the supplementary text: the reference list is bulleted and not numbered; sometimes the citations are of the form (Orth et al., 2011) and sometimes they are numbered.

Response: All references are now of the form Orth et al., 2011.

5. The authors use several times the word "predictability" where they seem to mean "predictive power" - e.g., supp text 4.1: "the predictability of the model increases proportionally to the correlation between..."

Response: We have replaced "predictability" by "predictive power" in the main text and suppl. methods.

6. The authors misspell numerous times the word "were" as "where" - e.g., in the Discussion section, 2nd paragraph: "interactions that where experimentally tested". This occurs both in the main text and in the supp text (supp text 5.1: "lower bounds on B where set to...")

Response: Corrected throughout the text.

7. The proper reference to FBA (both in main text and in the supp) is not Orth et al., MSB 2011. The authors should cite the original Varma & Palsson 1993 papers, and possibly the review in Orth et al., Nat Biotech 2010.

Response: We believe that the citation to Orth et al., MSB 2011 is appropriate, since this is the last FBA model that we used in our integrative model. We have additionally cited the suggested references in the Suppl. Material.

8. Supp text 4.4.1: mixed word order "to predict global the gene expression".

9. Incidentally, the authors could mention when discussing EBA that the approach of minimizing the change to the WT state following a perturbation is widely used in genome-scale modeling - this is the rationale of the MOMA method, Segre et al., PNAS 2002.

10. Supp text 4.4.1: shouldn't it be y_{TF} rather than y_g in the left-hand side of equation 16?

11. Supp text 4.4.2: shouldn't Z_i in equations 17-19 be replaced with 3?

12. Supp text 4.4.2: "we added a second criterion to characterize well-predicted arrays as those in which P was higher than a threshold" - what is P in this case? Do the authors mean the PCC?

13. Supp text 5.3: equation 22 introduces the parameter n, but I could not find where its value is given.

Response: All those comments were corrected throughout the text.

References:

1. Orth JD, et al (2011) A comprehensive genome-scale reconstruction of Escherichia coli metabolism--2011. *Mol Syst Biol* 7: 535.
2. O'Brien EJ, Lerman JA, Chang RL, Hyduke DR & Palsson BO (2013) Genome-scale models of metabolism and gene expression extend and refine growth phenotype prediction. *Mol Syst Biol* 9: 693.
3. Adadi R, et al (2012) Prediction of microbial growth rate versus biomass yield by a metabolic network with kinetic parameters. *PLoS Comp Biol* 8(7): e1002575.
4. Beg QK, et al (2007) Intracellular crowding defines the mode and sequence of substrate uptake by *Escherichia coli* and constrains its metabolic activity. *Proc Natl Acad Sci U S A* 104(31): 12663-12668.
5. Karr JR, Sanghvi JC, Jacobs JM, Macklin DN & Covert MW (2012) A whole cell model of mycoplasma genitalium elucidates mechanisms of bacterial replication. *Cell* 102(3): 731a.

Thank you again for submitting your revised work to Molecular Systems Biology. We are now satisfied with the modifications made and will be able to accept your manuscript for publication in Molecular Systems Biology pending the following minor points:

- Please change the style of the references to the MSB style.
- Please include a README file in the zip archive of Data Set 1, to explain what is the content of each file and to make explicit what are the columns and rows of the EcoMAC matrix.
- Please note that we have moved the Cytoscape files to 'source data files' associated with their respective figure panels, so that users can directly download them from the figure.